# A two-parameter learnable Logmoid Activation Unit

**Xue-Mei Zhou, Ling-Fang Li, Xing-Zhou Zheng**
School of Information Science and Technology
Southwest Jiaotong University
Chengdu 610031, China
`zhouxuemei0615@126.com,llf_hwj@163.com, zxzdyx123@126.com`

**Ming-Xing Luo**
School of Information Science and Technology
Southwest Jiaotong University
Chengdu 610031, China

**CSNMT Int. Coop. Res. Centre (MoST)**
Southwest Jiaotong University
Chengdu 610031, P.R. China
`mxluo@swjtu.edu.cn`

## ABSTRACT

A novel learnable Logmoid Activation Unit (LAU) is proposed as, $f(x) = x \ln(1+\alpha\text{Sigmoid}(\beta x))$, by parameterizing Sigmoid with two hyper-parameters $\alpha$ and $\beta$ that are optimized by the back-propagation algorithm. The end-to-end deep neural networks with learnable LAUs can increase the predictive performances beyond well-known activation functions for different tasks.

## 1 INTRODUCTION

Activation functions are generally classified into linear, nonlinear monotonic and nonlinear non-monotonic functions. Although linear functions have been widely used in early results, they are useless in practical applications. These problems are further addressed by using nonlinear monotonic functions such as Sigmoid, Tanh and ReLU families. While small derivatives may cause the gradient to disappear (Klambauer et al., 2017). Other characters such as the educed saturation, sparsity, efficiency, and ease of use will be explored.

Good activation function is still an open question. One is to design new activation functions by combining different units, such as Mish (Misra, 2019) and TanhExp (Liu & Di, 2020). The other is to parameterize some well-known activation functions Biswas et al. (2020); Zhou et al. (2020), which may show better performance beyond parameter-free functions. We proposed a learnable activation unit with new family of activation functions by parameterizing Logmoid with fewer trainable parameters for each network layer. This allows better simulations than others.

## 2 METHODS

The experiments simulate new activation function for different tasks based on the Logmoid Activation Unit (LAU) as

$$f(x; \alpha, \beta) = x \ln(1 + \alpha\sigma(\beta x)) \tag{1}$$

where $\alpha$ and $\beta$ are trainable hyper-parameters being optimized end-to-end to find a good one at each layer automatically, and $\sigma$ refers to Sigmoid=$\frac{1}{1+e^{-x}}$.

The parameter $\alpha$ basically determines the participation of the Sigmoid in the activation function. A very high of $\alpha$ may lead to a shape that is nowhere close to a ReLU-like function. The parameter $\beta$

controls the non-monotonic "bump" when $x < 0$. The hyper-parameters $1 \leqslant \alpha \leqslant 5$ and $1 \leqslant \beta \leqslant 5$ are for practical applications, and experiments on CIFAR-10 show that the Top-1 accuracies decrease faster when $-1 < \beta < 1$ and $\alpha = 1$. Therefore, we start with value 1 for both $\alpha$ and $\beta$ and train them with the same learning rate as used for the rest of the network, We name it Logmoid-1.

The output landscapes of a five-layer fully connected network in Figure 1 show that, While ReLU has lot of sharp transitions, Logmoid-1 gets a continuous and fluent transition shape. The landscapes were generated by passing in the co-ordinates to a five-layered randomly initialized neural network which outputs the corresponding scalar magnitude. Smoother output landscapes suggest smooth loss landscapes (Li et al., 2018), which help in easier optimization and better generalization.

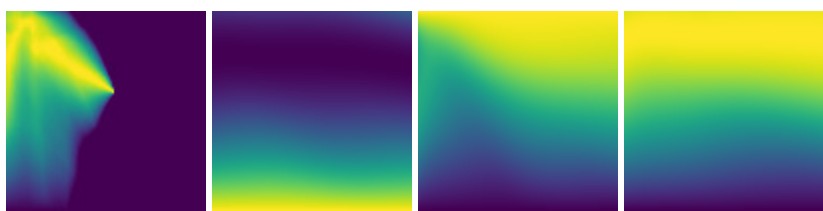

Figure 1: The output landscapes of ReLU, Swish, TanhExp and Logmoid-1.

By replacing the activation of the single hidden layer feed-forward neural network with Logmoid-1, the quasi-interpolation neural network operator $\overline{F}_{Logmoid-1}(f, x)$ is constructed, and it is theoretically proved that any continuous function $f(x)$ on the closed interval can be consistently approximated by the network operator, as shown in Theorem 1. The numerical experiment visualized in Figure 2 shows that the approximation error of Logmoid-1 is smaller than that of Swish.

**Theorem 1** *Let $\mathcal{C}_B(\mathbb{R})$ be the set of continuous and bounded function on $\mathbb{R}$. For any function $f \in \mathcal{C}_B(\mathbb{R})$, $n \in \mathbb{N}$, $0 < \alpha \leq 1$, $\gamma > 0$ we have*

$$\left| f(x) - \overline{F}_{Logmoid-1}(f, x) \right| \leq w\left( f, \frac{1}{n^\alpha} \right) + 8\gamma e^{-\frac{1}{2}n^{1-\alpha}} \|f\|_\infty$$

where $\| \cdot \|_\infty$ denotes the uniform norm.

## 3 EXPERIMENTS AND RESULTS

LAU networks are feed-forward networks including convolutional architectures with pooling layers, each LAU layer adds 2 additional parameters. Experiments are performed with Logmoid-1, ReLU, Swish, TanhExp, and ACONC (Ma et al., 2021) using different standard networks.

LAU always matches or outperforms the best performance on FMNIST, compared to the suboptimal results on VGG-8, LAU improved by 1.6 % Top-1 accuracy. Some learnable activations are smooth versions of Logmoid-1 by controlling hyper-parameters $\alpha$ and $\beta$. This shows LAU is a learnable activation function without requirement of additional suboptimal experiments. LAU is superior to the baseline activation functions in most cases on CIFAR-10 and CIFAR-100.

For the object detection by combing Mask R-CNN (He et al., 2017) with Swin Transformer (Liu et al., 2021), LAU improved the mAP@0.5 by 2.9% and 1.2 % compared with ReLU and ACONC. For a quick comparison, we randomly choose 200 classes from ImageNet, named as ImageNet-200. LAU gets 4.6% and 1.64 % Top-1 improvements over ReLU and ACONC on MobileNet-V3. Figure 4 shows that LAU gets a faster learning speed compared to other baseline activation functions.

## 4 CONCLUSIONS

The present novel learnable Logmoid Activation Unit is initialized using Logmoid-1, trainable in an end-to-end fashion. One can replace standard activation functions with LAU in any neural networks. This kind of functions may be potentially applied to approximate any continuous function. Simulations shows LAU has the best performance across all activation functions and architectures. Further work will be performed to apply LAU to other related tasks.

URM STATEMENT

The authors acknowledge that at least one key author of this work meets the URM criteria of the ICLR 2023 Tiny Papers Track.

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

# A  THEOREM 1 PROOF

Let $\mathcal{C}[-1, 1]$ be the space of continuous functions defined on $[-1, 1]$. For any function $f \in \mathcal{C}[a, b]$, the modulus of its continuity is defined by

$$w(f, \delta) := \max_{\substack{a \leq x, y \leq b \\ |x-y| \leq \delta}} |f(x) - f(y)|. \tag{2}$$

This modulus is used for measuring its approximation error and smoothness.

Now, construct a bell-shaped function $\Psi(x)$ using Logmoid-1,

$$\Psi(x) = \begin{cases} \psi(x + T_0) + 2|Y_1|, & \text{if } 0 \leq x < T_1 - T_0; \\ -\psi(x + T_0), & \text{if } x > T_1 - T_0; \\ \Psi(-x), & \text{if } x < 0. \end{cases} \tag{3}$$

where $T_0 = -0.2536$, $T_1 = 3.5025$ are the extreme points, and $Y_1 = -0.0181$ is the value of the $\psi(x)$ at point $T_1$, $\psi(x)$ is eq. 4:

$$\psi(x) \quad = \quad g(x + 1) - 2g(x) + g(x - 1) \tag{4}$$

where $g(x)$ is Logmoid-1.

Now, we define $\Gamma(x)$ as

$$\Gamma(x) = \begin{cases} e^{-\frac{1}{2}x - \frac{1}{2}}, & \text{if } x > 0; \\ e^{\frac{1}{2}x - \frac{1}{2}}, & \text{if } x \leq 0. \end{cases}$$

The bell-shaped functions $\Psi(x)$ satisfies the following properties.

(1) $\Psi(x) > 0$ for any $x \in \mathbb{R}$;

(2) $\Psi(x)$ is decreasing on $(0, +\infty)$, and increasing on $(-\infty, 0]$;

(3) $\Psi(x) \leq \Gamma(x)$ for any $x \in \mathbb{R}$.

A simple computation gives

$$0 < \sum_{k=-\infty}^{\infty} \Psi(x - k) \leq \int_{-\infty}^{\infty} \Psi(x) \, dx = 1.1918 := \xi \tag{5}$$

So, there exists a constant $\gamma > 0$, such that

$$\gamma \sum_{k=-\infty}^{\infty} \Psi(x - k) = 1. \tag{6}$$

Let $\Phi(x) = \gamma \Psi(x)$, $\Phi(x)$ has similar properties like $\Phi(x)$.

Then, let $\mathcal{C}_B(\mathbb{R})$ be the set of continuous and bounded function on $\mathbb{R}$. For any $f \in \mathcal{C}_B(\mathbb{R})$, construct FNN operator as

$$\overline{F}_{Logmoid-1}(f, x) := \sum_{n=-\infty}^{\infty} f\left(\frac{k}{n}\right) \Phi(nx - k). \tag{7}$$

So, we have

$$
\begin{aligned}
\left| f(x) - \overline{F}_{Logmoid-1}(f,x) \right| &= \left| f(x) \sum_{k=-\infty}^{\infty} \Phi(nx-k) - \sum_{k=-\infty}^{\infty} f\left(\frac{k}{n}\right) \Phi(nx-k) \right| \\
&\leq \sum_{k=-\infty}^{\infty} \left| f(x) - f\left(\frac{k}{n}\right) \right| \Phi(nx-k) \quad (8) \\
&= \sum_{k:\left|x-\frac{k}{n}\right|\leq\frac{1}{n^\alpha}} \left| f(x) - f\left(\frac{k}{n}\right) \right| \Phi(nx-k) \\
&\quad + \sum_{k:\left|x-\frac{k}{n}\right|>\frac{1}{n^\alpha}} \left| f(x) - f\left(\frac{k}{n}\right) \right| \Phi(nx-k) \\
&\leq w\left(f, \frac{1}{n^\alpha}\right) \sum_{k=-\infty}^{\infty} \Phi(nx-k) \quad (9) \\
&\quad + 2\|f\|_\infty \sum_{k:\left|x-\frac{k}{n}\right|>\frac{1}{n^\alpha}} \Phi(nx-k) \\
&\leq w\left(f, \frac{1}{n^\alpha}\right) + 2\|f\|_\infty \sum_{k:|nx-k|>n^{1-\alpha}} \Phi(nx-k). \quad (10)
\end{aligned}
$$

Here, we have used the triangle inequality for eq. 8, used eq. 2 to get eq. 9, used eq. 6 and triangle inequality to get eq. 9.

Since $\Phi(x)$ is strictly decreasing for any $x > 0$, it following that

$$
\begin{aligned}
\sum_{k:|nx-k|>n^{1-\alpha}} \Phi(nx-k) \leq 2\int_{n^{1-\alpha}-1}^{\infty} \Phi(x)dx &\leq 2\gamma \int_{n^{1-\alpha}-1}^{\infty} \Psi(x)dx \\
&\leq 2\gamma \int_{n^{1-\alpha}-1}^{\infty} e^{-\frac{1}{2}x-\frac{1}{2}}dx \\
&= 4\gamma e^{-\frac{1}{2}n^{1-\alpha}}
\end{aligned}
$$

Therefore, we get our Theorem 1.

## B   EXPERIMENTAL RESULTS

We define FNN operator with Logmoid-1 as follow:

$$
\overline{F}_{Logmoid-1}(f,x) := \sum_{n=-\infty}^{\infty} f\left(\frac{k}{n}\right) \Psi(nx-k). \quad (11)
$$

and $\overline{F}_{Swish-1}(f,x)$ as follow:

$$
\overline{F}_{Swish}(f,x) := \sum_{n=-\infty}^{\infty} f\left(\frac{k}{n}\right) \Omega(nx-k). \quad (12)
$$

where $\Omega(x)$ is defined as:

$$
\Omega(x) = \begin{cases} \phi(x) + 2|Y_1|, & \text{if } -T_1 \leq x \leq T_1; \\ -\phi(x), & \text{if } |x| > T_1. \end{cases} \quad (13)
$$

where $T_1 = 3.5893$ is extreme point, $Y_1 = -0.034$ is the value of the $\phi(x)$ at point $T_1$. $\phi(x)$ is based on Swish function, which is

$$\phi(x) = Swish(x+1) - 2 * Swish(x) + Swish(x-1). \tag{14}$$

Below, Figure 2 shows the approximation of function $y = \sin(\pi x)$ on [-5,5] using the FNN network operators $\overline{F}_{Logmoid-1}(f,x), \overline{F}_{Swish}(f,x)$ when $n = 30$.

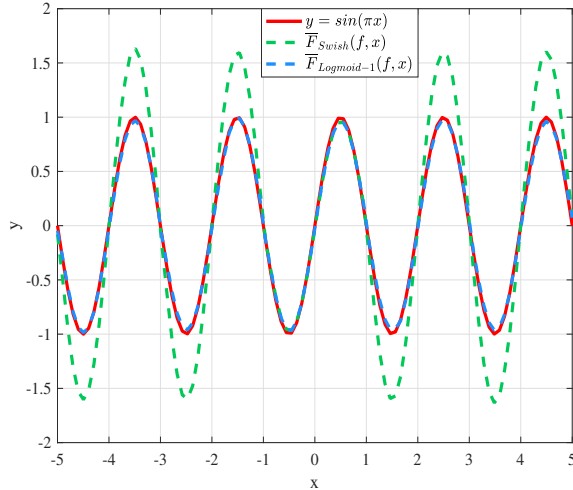

Figure 2: Uniform approximation of function $y = \sin(\pi x)$ on [-5,5] using the FNN network operators $\overline{F}_{Logmoid-1}(f,x), \overline{F}_{Swish}(f,x)$ for $n = 30$.

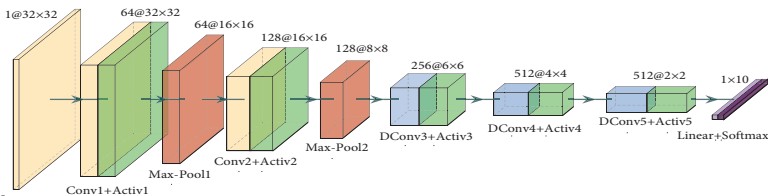

Figure 3: Schematic VGG-8-Dilated network. DConv denotes the dilated convolutional.

Table 1: Performance comparison of activation functions on CIFAR-10 (%).

| Model | ReLU | Swish | ACONC | TanhExp | Logmoid-1 | LAU |
|---|---|---|---|---|---|---|
| ResNet-20 | 90.96 | 91.08 | ∘91.16 | •**91.26** | 90.97 | ∘91.12 |
| MobileNet | 86.04 | 87.93 | ∘88.31 | 88.24 | 87.91 | •**88.53** |
| MobileNet-V2 | 84.92 | 85.87 | ∘86.54 | 85.83 | 86.39 | •**87.95** |
| ShuffleNet | 87.55 | 88.87 | 88.95 | ∘89.2 | 87.22 | •**89.91** |
| ShuffleNet-V2 | 87.42 | 87.62 | ∘87.64 | 87.38 | 87.18 | •**88.97** |
| SqueezeNet | •**88.50** | 87.04 | ∘88.35 | 87.86 | 87.63 | ∘88.22 |
| SeNet-18 | 88.81 | 90.53 | 90.37 | 90.19 | ∘90.95 | •**91.43** |
| EfficientNet-B0 | 76.17 | 77.35 | ∘78.55 | 77.19 | 78.23 | •**82.52** |
| TinyNet | 76.04 | ∘79.86 | 79.04 | 79.53 | 78.01 | •**80.21** |
| MicroNet | 62.79 | 67.16 | 68.48 | •**68.79** | 67.11 | ∘68.75 |

Table 2: Performance comparison of activation functions on CIFAR-100 (%).

| Model | ReLU | Swish | ACONC | TanhExp | Logmoid-1 | LAU |
|---|---|---|---|---|---|---|
| ResNet-20 | 66.16 | 67.73 | ●**67.86** | ○67.79 | 67.68 | ○67.77 |
| MobileNet | 59.40 | 62.11 | ●**62.35** | ○62.21 | 61.81 | ○62.05 |
| MobileNet-V2 | 59.40 | 61.32 | ○62.13 | 60.76 | 60.09 | ●**63.66** |
| ShuffleNet | 64.53 | 65.45 | 66.17 | 66.7 | ○66.76 | ●**68.01** |
| ShuffleNet-V2 | 63.65 | 65.88 | ○65.94 | ●**66.18** | 65.69 | 64.73 |
| SqueezeNet | 62.64 | ○63.97 | 63.45 | 62.38 | 63.64 | ●**64.26** |
| SeNet-18 | 66.26 | ○67.85 | 67.78 | 67.68 | 67.45 | ●**68.96** |
| EfficientNet-B0 | 45.81 | 46.01 | ○48.26 | 47.52 | 47.08 | ●**50.38** |
| TinyNet | 43.74 | 46.09 | ○48.64 | 47.92 | 48.39 | ●**50.86** |
| MicroNet | 30.77 | 35.56 | 33.91 | 36.59 | ○36.72 | ●**37.25** |

The Top-1 test accuracy for multiple activation functions on ShuffleNet-V2 in ImageNet-200 is shown in Fig. 4.

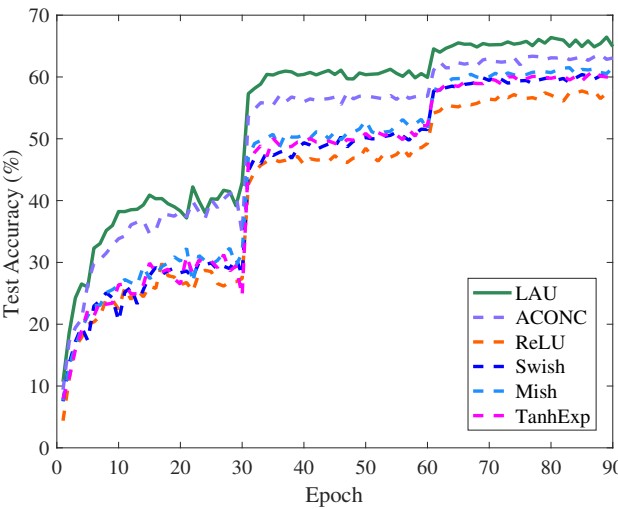

Figure 4: ShuffleNet-V2 Top-1 test accuracy for multiple activation functions in ImageNet-200.

Table 3: Performance comparison of activation functions on COCO 2017 (%).

| Activation | $AP^{box}$ | $AP_{50}^{box}$ | $AP_{75}^{box}$ | $AP^{mask}$ | $AP_{50}^{mask}$ | $AP_{75}^{mask}$ |
|---|---|---|---|---|---|---|
| ReLU | 45.8 | 64.6 | 50.3 | 41.2 | 63.3 | 44.7 |
| ACONC | 47.4 | 66.3 | 50.9 | 42.4 | 65.2 | 45.5 |
| LAU | 48.7 | 67.5 | 52.1 | 43.5 | 66.8 | 46.9 |

