# OpenReview forum: "A two-parameter learnable Logmoid Activation Unit"
_ICLR.cc/2023/TinyPapers — Submitted to Tiny Papers @ ICLR 2023_

### Official Review · Reviewer_AmHd · 2023-03-28

**Confidence:** 4

**Summary Of Contributions:**

This work introduces Logmoid, an activation function that combines the self-gating property of Swish with Sigmoid. It contains trainable parameters allowing it to adapt to the datasets being trained on. Shows promising results (for the most part) on CIFAR-10 across popular architectures.

**Rating:**

High Potential (HP): a submission which meets the reviewing criteria and has potential to make an impact on the field

**Strengths And Weaknesses:**

Hello authors, thank you for choosing to participate in Tiny Papers this year! Here are some comments:

**STRENGTHS**
- Paper introduces Logmoid / LAU that possesses nice properties and is learnable – simple method that seems to work well in the context
- Competitive performance on CIFAR-10 across popular vision architectures, demonstrating high applicability
- Clear coverage of how it differs from existing activation functions and why it possibly works well.

**WEAKNESSES*
- The task chosen (CIFAR-10) may not be the best benchmark to demonstrate. Good to have a few more difficult tasks (see possible improvements below)

**Suggested Changes:**

Good work on the paper! Here are some pressing changes to solidify your paper:

**URGENT**
- Typo: "to" instead of "for" in Conclusion ("may be potentially applied **to** approximate any ...").
- Benchmark on difficult benchmarks from computer vision (eg: segmentation, object detection, etc.), NLP (eg: NMT, question-answering), Graph DL (eg: ZINC), or RL (eg: standard gym environments) to show robustness and applicability. This is just to make the argument more convincing that your activation function is worth looking into. Check out [https://paperswithcode.com/area/computer-vision](https://paperswithcode.com/area/computer-vision) for possible tasks.
- Explain what the landscape diagram means – what are the axes and how to best visually interpret why your activation is better than the others shown (relu, swish, tanhexp).
- Question: Can I understand what you mean by "accuracy decreases faster for -1 < beta < 1 ..." in the Methods section below Equation (1)? Was this a typo? Did you mean "_increases_ faster"?

---

### Official Review · Reviewer_ACUp · 2023-03-29

**Confidence:** 4

**Summary Of Contributions:**

This paper proposes a variation of the logmoid activation function that contains learnable parameters that dynamically change the activations as the neural network is trained. The authors show that different experiments to validate their proposed activation function using different CNN architectures

**Rating:**

Needs Clarification (NC): a submission which does not meet the reviewing criteria and needs clarification for its described problem or solution

**Strengths And Weaknesses:**

**Strengths**
- The authors provide experiments against ACON[1] and TanhExp, both novel activation functions that have been shown to perform well. [1] in particular, also learns when to activate, so it makes for a great competitor.

**Weaknesses**
- Very few details are provided regarding the function and the impact of the parameters. From what I can see, if one makes a derivation of the values for the proposed activation function, $\alpha$ plays a critical role as it essentially decides the activation value:
$ln(1+\alpha\sigma(\beta x)) \= ln(1+ \frac{\alpha}{1+ e^{-\beta x}}) \= ln(\frac{1+ e^{-\beta x} + \alpha}{1+ e^{-\beta x}}) \=  ln(1+ e^{-\beta x}  + \alpha) -  ln(1+ e^{-\beta x})$. It seems to me that $\alpha$ is a crucial parameter and has a lot of influence on the activation, since if $\alpha=0$, we're essentially looking at a zero-ed output. The impact of the parameters must be expanded in more than 3 lines of text.

- The experimental section is very messy, the authors mention tests on fully-connected networks and basic convolutional architectures but no results are provided. The authors provide experiments with different convolutional architectures, yet there are no results using simple more complicated networks such as Transformers. In general, there are no reports of how much the proposed activation improves convergence.

- There are absolutely no backing demonstrations or experiments behind the phrases "One can replace standard activation functions with LAU units in any neural networks. This kind of functions may be potentially applied for approximate any continuous function". We know that NNs are universal function approximators, but we also know that the activation function plays a big role in this statement. I believe it is something that must be shown, even empirically, by comparing it with ReLU which is well-studied under a vast class of problems.

- The paper is badly written and quite messy. Numerous phrases are interrupted midway and there are a lot of typos, which makes reading the paper difficult.

[1] Ma, Ningning, et al. "Activate or not: Learning customized activation." Proceedings of the IEEE/CVF Conference on Computer Vision and Pattern Recognition. 2021.

**Suggested Changes:**

I believe that the points listed under the weaknesses contain different feedback for the authors that they should implement. Firstly, a larger set of empirical studies must be performed to begin validating their claims, and a detailed study of the impact of the parameter values in the approximation capability of the network and how they change the results. Last, but not least, the writing of the paper is poor and must be fixed.

---

### Author Response · Authors · 2023-05-30
**Official Comment of Paper241 by xuemei zhou**

I wish to opt-in for archival.

---

### Meta-Review · Area_Chair_QgSA · 2023-04-10

**Recommendation:** Invite to present
**Confidence:** 4

**Metareview:**

This paper proposes a new logmoid activation function variant (LAU) with learnable parameters. Promising experimental results are shown to validate the new activation function. I think this paper is interesting. There are some pros and cons which are listed below.

Pros:
1. A new logmoid activation function (LAU) with good properties is proposed.
2. The authors demonstrate that LAU can outperform existing activation functions on some tasks.
3. The paper provides a clear coverage of the difference between LAU and other activation functions and discusses the possible reasons why it works well.

Cons:
1. The authors are suggested to provide a better explanation of the roles/intuitions of \alpha and \beta in LAU.
2. The authors are encouraged to test the effectiveness of LAU using more challenging benchmarks.
3. The writing can be improved a bit (such as some typos, figures, etc.).

**Summary:**

This tiny paper proposes a new activation function consisting of two parameters and shows some promising experimental results. However, some reviewers find it necessary to perform more comprehensive empirical studies and polish the paper writing.

**Reason For Not Giving A Higher Recommendation:**

1. The authors are suggested to provide a better explanation of the roles/intuitions of \alpha and \beta in LAU.
2. The authors are encouraged to test the effectiveness of LAU using more challenging benchmarks.
3. The writing can be improved a bit (such as some typos, figures, etc.).

**Reason For Not Giving A Lower Recommendation:**

1. A new logmoid activation function (LAU) with good properties is proposed.
2. The authors demonstrate that LAU can outperform existing activation functions on some tasks.
3. The paper provides a clear coverage of the difference between LAU and other activation functions and discusses the possible reasons why it works well.

---

### Decision · Program_Chairs · 2023-04-10

Invite to present